# EfficientDM: Efficient Quantization-Aware Fine-Tuning of Low-Bit Diffusion Models

**Yefei He[1]   Jing Liu[2]   Weijia Wu[1]   Hong Zhou[1]\*   Bohan Zhuang[2]\***

[1]Zhejiang University, China
[2]ZIP Lab, Monash University, Australia

## Abstract

Diffusion models have demonstrated remarkable capabilities in image synthesis and related generative tasks. Nevertheless, their practicality for low-latency real-world applications is constrained by substantial computational costs and latency issues. Quantization is a dominant way to compress and accelerate diffusion models, where post-training quantization (PTQ) and quantization-aware training (QAT) are two main approaches, each bearing its own properties. While PTQ exhibits efficiency in terms of both time and data usage, it may lead to diminished performance in low bit-width settings. On the other hand, QAT can help alleviate performance degradation but comes with substantial demands on computational and data resources. To capitalize on the advantages while avoiding their respective drawbacks, we introduce a data-free, quantization-aware and parameter-efficient fine-tuning framework for low-bit diffusion models, dubbed EfficientDM, to achieve QAT-level performance with PTQ-like efficiency. Specifically, we propose a quantization-aware variant of the low-rank adapter (QALoRA) that can be merged with model weights and jointly quantized to low bit-width. The fine-tuning process distills the denoising capabilities of the full-precision model into its quantized counterpart, eliminating the requirement for training data. To further enhance performance, we introduce scale-aware LoRA optimization to address ineffective learning of QALoRA due to variations in weight quantization scales across different layers. We also employ temporal learned step-size quantization to handle notable variations in activation distributions across denoising steps. Extensive experimental results demonstrate that our method significantly outperforms previous PTQ-based diffusion models while maintaining similar time and data efficiency. Specifically, there is only a marginal 0.05 sFID increase when quantizing both weights and activations of LDM-4 to 4-bit on ImageNet $256 \times 256$. Compared to QAT-based methods, our EfficientDM also boasts a $16.2\times$ faster quantization speed with comparable generation quality, rendering it a compelling choice for practical applications. Code is available at https://github.com/ThisisBillhe/EfficientDM.

## 1 Introduction

Diffusion models (DM) (Ho et al., 2022b; Dhariwal & Nichol, 2021; Rombach et al., 2022a; Ho et al., 2022a) have demonstrated remarkable capabilities in image generation and related tasks. Nonetheless, the iterative denoising process and the substantial computational overhead of the denoising model limit the efficiency of DM-based image generation. To expedite the image generation process, numerous methods (Bao et al., 2022; Song et al., 2021; Liu et al., 2022; Lu et al., 2022) have been explored to reduce the number of denoising iterations, effectively reducing the previously required thousands of iterations to mere dozens. However, the significant volume of parameters within the denoising model still demands a substantial computational burden for each denoising step, resulting in considerable latency, hindering the practical application of DM in real-world settings with latency and computational resource constraints.

---

\*Corresponding author. Email: `zhouhong_zju@zju.edu.cn`, `bohan.zhuang@gmail.com`

Model quantization, which compresses weights and activations from 32-bit floating-point values into lower-bit fixed-point formats, alleviating both memory and computational burdens. This effect can be increasingly pronounced as the bit-width decreases. For instance, leveraging Nvidia's CUTLASS (Kerr et al., 2017) implementation, an 8-bit model's inference speed can be $2.03\times$ faster than that of a full-precision (FP) model, and the acceleration ratio reaches $3.34\times$ for a 4-bit model. Therefore, it possesses substantial potential for significantly compressing and accelerating diffusion models, making them highly suitable for deployment on resource-constrained devices such as mobile phones.

Nevertheless, the challenges associated with low-bit quantization for diffusion models have not received adequate attention. Typically, model quantization can be executed through two predominant approaches: post-training quantization (PTQ) and training-aware quantization (QAT). PTQ calibrates the quantization parameters with a small calibration dataset, which is time- and data-efficient. However, they introduce substantial quantization errors at low bit-width. As illustrated in Figure 1 , when quantizing both weights and activations to 4-bit with the PTQ method (He et al., 2023a), diffusion models fail to maintain their denoising capabilities. In contrast, QAT methods (Krishnamoorthi, 2018; Esser et al., 2019) can recover performance losses at lower bit-width by fine-tuning the whole model. However, this approach requires significantly more time and computing resources compared to PTQ method (He et al., 2023a), as evidenced by a $2.6\times$ increase in GPU memory consumption (31.4GB vs. 11.7GB) and a $18.9\times$ longer execution time (54.5 GPU hours vs. 2.88 GPU hours) when fine-tuning LDM-4 (Rombach et al., 2022a) on ImageNet $256 \times 256$. Moreover, in some cases, it may be challenging or even impossible to obtain the original training dataset due to privacy or copyright concerns.

In this paper, we introduce a data-free and parameter-efficient fine-tuning framework for low-bit diffusion models, denoted as EfficientDM, which demonstrates the capability to achieve *QAT-level performance* while upholding *PTQ-level efficiency* during fine-tuning in terms of data and time. The foundation of our approach lies in a quantization-aware form of the low-rank adapter, as depicted in Figure 2b. This variant enables the joint quantization of LoRA weights with model weights, thereby obviating additional storage and calculations. Compared to previous QAT method (So et al., 2023), our fine-tuning process is executed in a data-free manner, accomplished by minimizing the mean squared error (MSE) between the estimated noise of full-precision denoising model and its quantized counterpart, as illustrated in Figure 2c, thus eliminating the need for the original training dataset. Due to quantization, the relationship between full-precision LoRA weights and quantized updated model weights becomes step-like with step size, *i.e.*, the quantization scale parameter. Effective updates are mostly observed in layers with small scales, while other layers with larger scales do not benefit. To address this, we introduce scale-aware LoRA optimization that adaptively adjusts the gradient scales of LoRA weights in different layers to ensure an effective optimization. Furthermore, we extend the learned step size quantization (LSQ) method (Esser et al., 2019) into the denoising temporal domain for

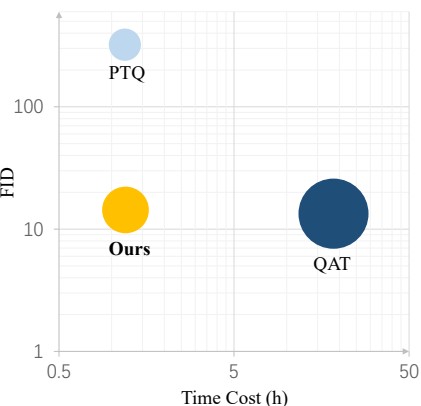

Figure 1: An overview of the efficiency-vs-quality tradeoff across various quantization approachs. Data is collected on LDM-8 (Rombach et al., 2022a) with 4-bit weights and activations on LSUN-Churches. The GPU memory consumption is visualized by circle size.

activations, effectively mitigating quantization errors due to varying activation distributions across time steps.

In summary, our contributions are as follows:

- We introduce EfficientDM, an efficient fine-tuning framework for low-bit diffusion models which can achieve QAT performance with the efficiency of PTQ. The framework is rooted in the quantization-aware form of low-rank adapters (QALoRA) and distills the denoising capabilities of full-precision models into their quantized counterparts.
- We propose scale-aware LoRA optimization to alleviate the ineffective learning of QALoRA resulting from substantial variations in weight quantization scales across different layers. We also

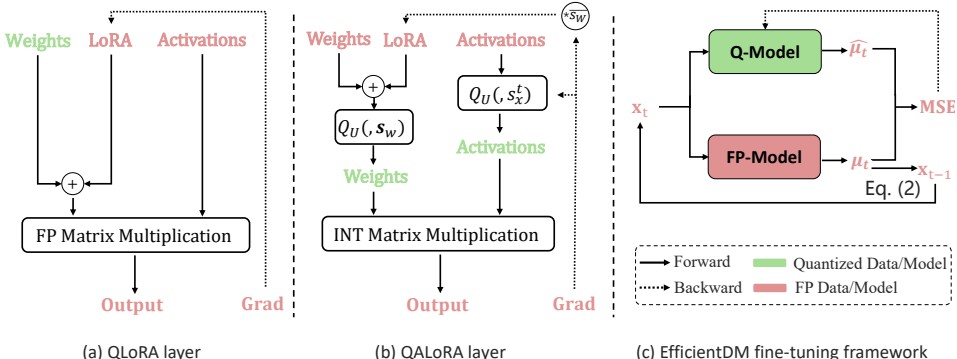

(a) QLoRA layer  (b) QALoRA layer  (c) EfficientDM fine-tuning framework

Figure 2: An overview of the proposed EfficientDM fine-tuning framework. Here, $\mathbf{s}_w$ and $s_x$ represent the learnable quantization scales for weights and activations, respectively. Compared to QLoRA layer, both updated weights and activations in our QALoRA are quantized to enable efficient bitwise operations during inference. Fine-tuning is performed by minimizing the mean squared error between the estimated noises of FP and quantized models.

> introduce TALSQ, an extension of activation learned step size quantization within the temporal domain to tackle the variation in activation distributions across denoising steps.
>
> • Extensive experiments on CIFAR-10, LSUN and ImageNet demonstrate that our EfficientDM reaches a new state-of-the-art performance for low-bit quantization of diffusion models.

## 2 RELATED WORK

**Model quantization.** Quantization is a widely employed technique for compressing and accelerating neural networks. Depending on whether fine-tuning of the model is necessitated, this method can be categorized into two approaches: post-training quantization (PTQ) (Nagel et al., 2020; Li et al., 2021; Wei et al., 2022; Lin et al., 2022) and quantization-aware training (QAT) (Esser et al., 2019; Gong et al., 2019; Nagel et al., 2022; Louizos et al., 2019; Jacob et al., 2018; Zhuang et al., 2018; He et al., 2023b). PTQ does not involve fine-tuning the model's weights and only requires a small dataset to calibrate the quantization parameters. This approach is fast and data-efficient but may result in suboptimal performance especially when employing low bit-widths. Recent advancements in reconstruction-based PTQ methods (Li et al., 2021; Wei et al., 2022) utilize second-order error analysis and gradient descent algorithms to optimize quantization parameters. Such methods demonstrate robust performance, which can perform well even at 4-bit on image classification tasks. On the other hand, QAT entails fine-tuning the model weights to achieve low-bit quantization while minimizing performance degradation. Nevertheless, the substantial data prerequisites and high computational overhead render QAT much slower than PTQ. For instance, when applied to ResNet-18 model for the ImageNet classification task, it is reported to be $240\times$ slower than the PTQ method (Li et al., 2021). This hindrance poses challenges to its widespread adoption, particularly for models with massive parameters.

Given the demand to compress diffusion models and accelerate their sampling, quantization is considered an effective approach. Currently, most quantization work on diffusion models has focused on PTQ (Shang et al., 2023; Li et al., 2023; He et al., 2023a). Since diffusion models can generate samples from random Gaussian noise, these methods are conducted in a data-free manner, with the calibration sets collected from the denoising process. For example, PTQ4DM (Shang et al., 2023) and Q-Diffusion (Li et al., 2023) apply reconstruction-based PTQ methods to diffusion models. PTQD (He et al., 2023a) further decomposes quantization noise and fuses it with diffusion noise. Nevertheless, these methods experience significant performance degradation at 4-bit or lower bit-widths. Recent work TDQ (So et al., 2023) fine-tunes diffusion models in a QAT manner and employs additional MLP modules to estimate quantization scales for each step. However, it requires the original dataset and training the entire quantized model with these extra MLP modules, incurring significant costs. In contrast, our approach introduces only a few trainable quantization scales per layer and can attain QAT-level performance with PTQ-level efficiency.

**Parameter-efficient fine-tuning.** Parameter-Efficient Fine-Tuning (PEFT) has emerged as a potent alternative to full fine-tuning, which tunes a small subset of parameters while keeping the vast ma-

jority frozen, to ease storage burden of large pretrained models. Noteworthy approaches within this realm include Prefix Tuning (Li & Liang, 2021; Liu et al., 2021), Prompt Tuning (Lester et al., 2021; Wang et al., 2022), and Low-rank adapters (LoRA) (Hu et al., 2022; Zhang et al., 2022), which have found extensive utility within large language models (LLMs).

Among these PEFT methods, LoRA (Hu et al., 2022) keeps the pre-trained weights fixed and learns a set of low-rank decomposition matrices as updates to the weights. This method substantially reduces the number of trainable parameters, and does not impose additional computational overhead due to re-parameterization. QLoRA (Dettmers et al., 2023) combines this technique with quantized LLMs, further curtailing memory requirements while mitigating performance degradation brought by quantization. However, its full-precision LoRA weights cannot be integrated with quantized model weights, leading to extra storage and computational costs for deployment. In contrast, we propose to merge LoRA weights with full-precision model weights and jointly quantize them to the target bit-width, enabling their seamless integration without incurring any additional overhead for deployment. Additionally, we identify and address the issue of ineffective learning of LoRA weights stemming from the variations in weight quantization scales, further enhancing the performance of low-bit diffusion models.

## 3 BACKGROUND

### 3.1 DIFFUSION MODELS

Diffusion models (Song et al., 2021; Ho et al., 2020) are a class of generative models that iteratively introduce noise to real data $\mathbf{x}_0$ through a forward process and generate high-quality samples via a reverse denoising process. Generally, the forward process is a Markov chain, which can be formulated as:

$$q(\mathbf{x}_t|\mathbf{x}_{t-1}) = \mathcal{N}(\mathbf{x}_t; \sqrt{1-\beta_t}\mathbf{x}_{t-1}, \beta_t\mathbf{I}), \tag{1}$$

where $\beta_t$ governs the magnitude of the introduced noise. The reverse process adheres to a similar form, where diffusion models approximate the distribution of $q(\mathbf{x}_{t-1}|\mathbf{x}_t)$ via variational inference by learning a Gaussian distribution:

$$p_\theta(\mathbf{x}_{t-1}|\mathbf{x}_t) = \mathcal{N}(\mathbf{x}_{t-1}; \boldsymbol{\mu}_\theta(\mathbf{x}_t, t), \boldsymbol{\Sigma}_\theta(\mathbf{x}_t, t)), \tag{2}$$

where $\boldsymbol{\mu}_\theta$ and $\boldsymbol{\Sigma}_\theta$ are two neural networks. When these neural networks are subjected to quantization, the resulting estimated statistics of the Gaussian distribution can be inaccurate, thereby compromising the efficacy of the denoising process and the overall quality of the generated samples. Our study is focused on introducing an efficient and powerful fine-tuning framework to narrow or even eliminate the performance gap between low-bit diffusion models and their full-precision counterparts.

### 3.2 MODEL QUANTIZATION

Model quantization represents model parameters and activations with low-precision integer values to reduce memory footprint and accelerate the inference. Given a floating-point vector $\mathbf{x}$, it can be uniformly quantized as follows:

$$\hat{\mathbf{x}} = \mathcal{Q}_U(\mathbf{x}, s) = \text{clip}(\lfloor \frac{\mathbf{x}}{s} \rceil, l, u) \cdot s. \tag{3}$$

Here, $\lfloor \cdot \rceil$ is the round operation, $s$ is the trainable quantization scale, and $l$ and $u$ are the lower and upper bound of quantization thresholds, which are determined by the target bit-width. To overcome the non-differentiability of the round operation during QAT, straight-through estimator (STE) is widely adopted to estimate the gradient:

$$\frac{\partial \mathcal{L}}{\partial \mathbf{x}} \approx \frac{\partial \mathcal{L}}{\partial \hat{\mathbf{x}}} \cdot \mathbf{1}_{l \leq \frac{\mathbf{x}}{s} \leq u}, \tag{4}$$

where $\mathbf{1}_A$ represents an indicator function that takes on the value 1 if the element is in the set $A$, and the value 0 vice versa.

## 4 EFFICIENT QUANTIZATION-AWARE FINE-TUNING OF DIFFUSION MODELS

In this section, we propose an efficient fine-tuning framework for diffusion models with both weights and activations quantized, possessing the efficiency of PTQ and the accuracy of QAT. The proposed framework, dubbed EfficientDM, is depicted in Figure 2. It consists of a quantization-aware low-rank adapter and a noise distillation strategy, delivering parameter-efficient and data-free fine-tuning. Moreover, it is also equipped with scale-aware LoRA optimization and a temporal-aware quantizer for improved performance. We elaborate each design as follows.

**Quantization-aware low-rank adapter.** Low-rank adapter (LoRA) fine-tuning constrains the update of the model parameters to possess a low intrinsic rank, denoted as $r$. Given a pretrained linear module $\mathbf{Y} = \mathbf{X}\mathbf{W}_0$, where $\mathbf{X} \in \mathbb{R}^{b \times c_{in}}$ and $\mathbf{W}_0 \in \mathbb{R}^{c_{in} \times c_{out}}$, with $b$ representing the batch size, $c_{in}$ and $c_{out}$ representing the number of input and output channels, respectively, LoRA fixes the original weights $\mathbf{W}_0$ and introduces updates as follows:

$$\mathbf{Y} = \mathbf{X}\mathbf{W}_0 + \mathbf{X}\mathbf{B}\mathbf{A}, \tag{5}$$

where $\mathbf{B} \in \mathbb{R}^{c_{in} \times r}$ and $\mathbf{A} \in \mathbb{R}^{r \times c_{out}}$ are the learnable two low-rank matrices with $r \ll \min(c_{in}, c_{out})$.

Nevertheless, this approach incurs limitations when both weights and activations are quantized, denoted by $\hat{\mathbf{W}}_0$ and $\hat{\mathbf{X}}$, respectively. In this case, the inner product between $\hat{\mathbf{W}}_0$ and $\hat{\mathbf{X}}$ can be efficiently implemented with bit-wise operations, whereas the operations involving $\mathbf{B}\mathbf{A}$ and $\hat{\mathbf{X}}$ are computationally expensive during inference as $\mathbf{B}\mathbf{A}$ is full-precision and has the same size as $\mathbf{W}_0$.

To address this, we propose Quantization-aware Low-rank Adapter (QALoRA), where the LoRA weights are first merged with FP model weights and then jointly quantized to the target bit-width, as depicted in Figure 2b. Formally, the QALoRA is defined as follows:

$$\mathbf{Y} = \mathcal{Q}_U(\mathbf{X}, s_x)\mathcal{Q}_U(\mathbf{W}_0 + \mathbf{B}\mathbf{A}, \mathbf{s}_w) = \hat{\mathbf{X}}\hat{\mathbf{W}}, \tag{6}$$

where $\mathbf{s}_w$ denotes the channel-wise quantization scale for weights and $s_x$ is the layer-wise quantization scale for activations. After the fine-tuning process, only quantized updated model weights $\hat{\mathbf{W}}$ need to be saved. Notably, our approach can be readily integrated with QLoRA (Dettmers et al., 2023) by substituting $\mathbf{W}_0$ with $\hat{\mathbf{W}}_0$ to further reduce memory footprint.

**Data-free fine-tuning for diffusion models.** Diffusion models require access to large and diverse datasets for effective training. Obtaining such datasets can be challenging due to their sheer size, privacy concerns, or copyright restrictions. To alleviate the dependency on the original dataset, we propose a data-free fine-tuning approach that distills the denoising capabilities of a full-precision model into its quantized counterpart. Specifically, we input the same noise $\mathbf{x}_t$ to both FP and quantized denoising models at denoising step $t$ and minimize the mean squared error (MSE) between their denoising results:

$$\mathcal{L}_t = \|\boldsymbol{\mu}_\theta(\mathbf{x}_t, t) - \hat{\boldsymbol{\mu}}_\theta(\mathbf{x}_t, t)\|^2, \tag{7}$$

where $\boldsymbol{\mu}_\theta(\mathbf{x}_t, t)$ and $\hat{\boldsymbol{\mu}}_\theta(\mathbf{x}_t, t)$ denote the denoising results of the FP and quantized models for the denoising step $t$, respectively. The input data $\mathbf{x}_t$ is obtained by denoising random Gaussian noise $\mathbf{x}_T \sim \mathcal{N}(0, 1)$ with FP model iteratively for $T - t$ steps, as illustrated in Figure 2c.

To facilitate the training of QALoRA, we further address the following technical challenges:

**Variation of weight quantization scales across layers.** As demonstrated in Eq. (6), due to the quantization process, the relationship between full-precision LoRA weights $\mathbf{B}\mathbf{A}$ and quantized updated weights $\hat{\mathbf{W}}$ follows a step function, where the step size is exactly equal to the quantization scale $\mathbf{s}_w$, and $\mathbf{W}_0$ serves as a fixed offset. This relationship is visually presented in Figure 3a. Consequently, while the full-precision LoRA weights are continuously optimized during the fine-tuning process, they need to be large enough to update the quantized model weights, otherwise they will be diminished by the round operation within the quantization process, as referred to Eq. (3) and (6). As shown in Figure 3b and "Scale-agnostic training" in Figure 3c, fine-tuning the LoRA weights with limited iterations only yields effective updates for a few layers with relatively small scales. For other layers, their quantization scales are too substantial for LoRA weights to take effect due to the round operation. Alternative approaches, such as directly amplifying the learning rate, may impede the convergence process.

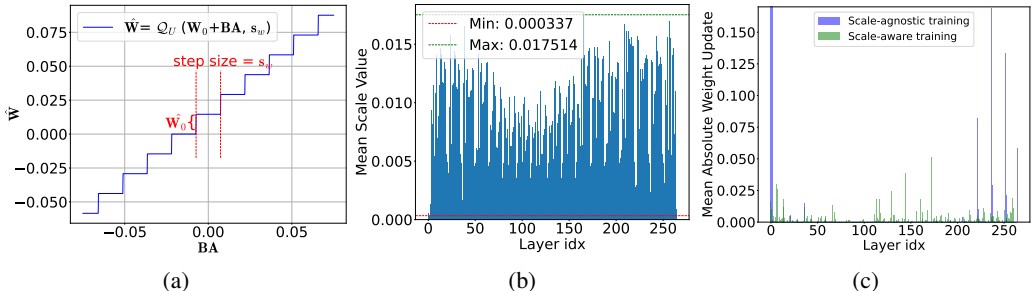

(a)                                        (b)                                        (c)

Figure 3: The motivation and effect of scale-aware LoRA optimization. Data is collected from the 4-bit LDM-4 model. **(a):** Due to step-like relationship between $\mathbf{BA}$ and $\hat{\mathbf{W}}$, $\mathbf{BA}$ needs to be large enough to update model weights. Data is collected from the first channel in the $12^{th}$ layer. **(b):** Significant disparity in weight quantization scales across layers. **(c):** Mean absolute value of quantized weight updates ($\hat{\mathbf{W}}$-$\hat{\mathbf{W}}_0$) for each layer. Most of them are zero under scale-agnostic training, indicating full-precision LoRA weights are too small to update quantized model weights. The proposed scale-aware LoRA optimization facilitates a more equitable distribution of quantized weight updates across layers.

To facilitate the optimization of LoRA weights, we consider the ratio of

$$R = \frac{\nabla_{\mathbf{BA}}\mathcal{L}}{\overline{\mathbf{s}}_w} \tag{8}$$

should be roughly consistent in each layer, where $\overline{\mathbf{s}}_w$ represents the averaged weight quantization scale across channels. This can be achieved by simply multiplying the gradient of LoRA weights by this average weight quantization scale during the back-propagation phase. As shown in Figure 3c, optimizing LoRA in the scale-aware approach enables effective training across the majority of layers.

**Variation of activation distribution across steps.** Previous research on diffusion models (Shang et al., 2023; Li et al., 2023; So et al., 2023) has identified a pronounced variability in activation distributions at different time steps, which is also presented in Appendix C. This variability poses substantial challenges to the quantization of diffusion models. Notably, existing methods proposed to address this issue aim to either find a set of quantization parameters applicable to all time steps (Shang et al., 2023) or employ additional MLP module to estimate quantization parameters for each individual time step and layer (So et al., 2023), which can be either suboptimal or cumbersome.

Inspired by LSQ (Esser et al., 2019), a technique where quantization scales are optimized alongside other trainable parameters through the gradient descent algorithm, we allocate temporal-aware quantization scales for activations and optimize them individually for each step, which we refer to as Temporal Activation LSQ (TALSQ):

$$S_x = \left\{ s_x^0, s_x^1, \ldots, s_x^{T-1} \right\}, \tag{9}$$

where $T$ is the number of denoising steps for the fine-tuning. It is noteworthy that recent advancements in efficient samplers have significantly reduced the number of sampling steps. Therefore, TALSQ introduces only a few trainable parameters for a single layer, which is negligible even compared to LoRA weights (which generally have thousands of parameters per layer). After fine-tuning, we interpolate the learned temporal quantization scales to deal with the gap of sampling steps between fine-tuning and inference.

## 5 EXPERIMENTS

### 5.1 IMPLEMENTATION DETAILS

**Models and metrics.** To verify the effectiveness of the proposed method, we evaluate it with two widely adopted network structures: DDIM (Song et al., 2021) and LDM (Rombach et al., 2022b). For experiments with DDIM, we evaluate it on CIFAR-10 dataset (Krizhevsky & Hinton, 2009). Experiments with LDM are conducted on two standard benchmarks: ImageNet (Deng et al., 2009) and LSUN (Yu et al., 2015). The performance of diffusion models is evaluated with Inception Score (IS), Fréchet Inception Distance (FID) (Heusel et al., 2017). For experiments on ImageNet datasets,

we also report sFID (Salimans et al., 2016) and Precision for reference. Results are obtained by sampling $50,000$ images and evaluating them with ADM's TensorFlow evaluation suite (Dhariwal & Nichol, 2021). The rank of the adapter is set to 32 in our experiments. For the proposed EfficientDM framework, we fine-tune LoRA weights and quantization parameters for 16K iterations with a batchsize of 4 on LDM models and 64 on DDIM models, respectively. The number of denoising steps for the fine-tuning is set to 100. We employ Adam (Kingma & Ba, 2014) optimizers with a learning rate of $5e^{-4}$.

**Quantization settings.** The notion 'W$x$A$y$' is employed to represent the bit-widths of weights 'W' and activations 'A'. We use channel-wise quantization for weights and layer-wise quantization for activations, as is a common practice. The input embedding and output layers in the model employ a fixed W8A8 quantization, whereas other convolutional and fully-connected layers are quantized to the target bit-width with a QALoRA module.

## 5.2 MAIN RESULTS

### 5.2.1 EVALUATION OF UNCONDITIONAL GENERATION

We begin our evaluation by performing unconditional generation on CIFAR-10 dataset and compare our results against well-established quantization methods, including both PTQ-based (Shang et al., 2023; Li et al., 2023) and QAT-based (Esser et al., 2019; So et al., 2023) contenders. The results are presented in Table 1. Recent PTQ-based method (Li et al., 2023) performs well under W8A8 precision but exhibits performance degradation under W4A8 precision. Recent QAT-based method (So et al., 2023) surpasses all PTQ-based approaches under W4A8 precision but necessitates access to the original training dataset and incurs a substantial $17.8\times$ increase in training time. In contrast, our proposed EfficientDM demonstrates superior performance even compared to previous QAT methods under both W8A8 and W4A8 precisions, while incurring significantly lower time costs. Notably, under W4A8 precision, we achieve an FID of $3.80$, outperforming QAT-based method TDQ (So et al., 2023) by $0.33$.

Additional experimental results on LSUN dataset can be found in Appendix A.

Table 1: Performance comparisons of quantized diffusion models on CIFAR-10 $32 \times 32$. Results are obtained by DDIM sampler with 100 steps.

| Method | Bit-width (W/A) | Training data | GPU Time (hours) | Model Size (MB) | IS↑ | FID↓ |
|---|---|---|---|---|---|---|
| FP | 32/32 | 50K | - | 136.4 | 9.12 | 4.14 |
| PTQ4DM | 8/8 | 0 | 0.95 | 34.26 | 9.31 | 14.18 |
| Q-Diffusion | 8/8 | 0 | 0.95 | 34.26 | 9.48 | 3.75 |
| LSQ | 8/8 | 50K | 13.89 | 34.26 | **9.62** | 3.87 |
| TDQ | 8/8 | 50K | 16.99 | 34.30 | 9.58 | 3.77 |
| Ours | 8/8 | 0 | 0.97 | 34.30 | 9.38 | **3.75** |
| PTQ4DM | 4/8 | 0 | 0.95 | 17.22 | 9.31 | 10.12 |
| Q-Diffusion | 4/8 | 0 | 0.95 | 17.22 | 9.12 | 4.93 |
| LSQ | 4/8 | 50K | 13.89 | 17.22 | 9.38 | 4.53 |
| TDQ | 4/8 | 50K | 16.99 | 17.26 | **9.59** | 4.13 |
| Ours | 4/8 | 0 | 0.97 | 17.26 | 9.41 | **3.80** |

### 5.2.2 EVALUATION OF CONDITIONAL GENERATION ON IMAGENET $256 \times 256$

We also evaluate the performance of our method over LDM-4 on large-scale ImageNet $256 \times 256$ dataset for class-conditional image generation, as presented in Table 2. Experiments are conducted with three distinct samplers, including DDIM (Song et al., 2021), PLMS (Liu et al., 2022) and DPM-Solver (Lu et al., 2022). Importantly, we offer a comprehensive evaluation based on four metrics (IS, FID, sFID, and Precision), which should be collectively considered to ensure a robust assessment of image quality. In W8A8 quantization, recent PTQ-based methods (Li et al., 2023; He et al., 2023a) exhibit performance levels with negligible degradation. However, when the bit-width drops to W4A8, performance degradation becomes evident for them. For instance, when employing the DDIM sampler, Q-Diffusion experiences a decline of 27.8 in IS and a decrease of $2.6\%$ in precision. The introduction of EfficientDM yields a significant recovery in performance, with a merely 2.4 IS decrease compared with FP method. As the bit-width decreased, the performance enhancements

of EfficientDM became more pronounced. In the case of W4A4 quantization, none of the diffusion models quantized with PTQ-based methods is able to denoise images, regardless of the sampler used. In contrast, models quantized by our approach manage to maintain an exceptionally low FID of 6.17 and a sFID of 7.75 for the DDIM sampler, respectively. Leveraging the capabilities of EfficientDM, we push the quantization of diffusion model weights to 2-bit for the first time, resulting in a marginal increase of less than 1 in the sFID metric when using the DDIM sampler. Moreover, we present the real-time speed up of EfficientDM over LDM-4 in Appendix B.

Table 2: Performance comparisons of fully-quantized LDM-4 models on ImageNet $256 \times 256$. 'N/A' denotes failed image generation.

| Model-Sampler | Method | Bit-width (W/A) | IS↑ | FID↓ | sFID↓ | Precision↑ (%) |
|---|---|---|---|---|---|---|
| | FP | 32/32 | 364.73 | 11.28 | 7.70 | 93.66 |
| | Q-Diffusion | 8/8 | 350.93 | 10.60 | 9.29 | 92.46 |
| | PTQD | 8/8 | 359.78 | **10.05** | 9.01 | 93.00 |
| | Ours | 8/8 | **362.34** | 11.38 | **8.04** | **93.77** |
| | Q-Diffusion | 4/8 | 336.80 | 9.29 | 9.29 | 91.06 |
| | PTQD | 4/8 | 344.72 | **8.74** | 7.98 | 91.69 |
| LDM-4 — DDIM (20 steps) | Ours | 4/8 | **353.83** | 9.93 | **7.34** | **93.10** |
| | Q-Diffusion | 4/4 | N/A | N/A | N/A | N/A |
| | PTQD | 4/4 | N/A | N/A | N/A | N/A |
| | Ours | 4/4 | **250.90** | **6.17** | **7.75** | **86.02** |
| | Q-Diffusion | 2/8 | 49.08 | 43.36 | 17.15 | 43.18 |
| | PTQD | 2/8 | 53.36 | 39.37 | 15.14 | 45.89 |
| | Ours | 2/8 | **175.03** | **7.60** | **8.12** | **78.90** |
| | FP | 32/32 | 379.19 | 11.71 | 6.08 | 94.22 |
| | Q-Diffusion | 8/8 | 373.49 | 11.25 | 5.75 | 93.84 |
| | PTQD | 8/8 | 374.50 | **11.05** | **5.45** | 94.00 |
| | Ours | 8/8 | **376.06** | 11.78 | 6.21 | **94.27** |
| | Q-Diffusion | 4/8 | 358.13 | 9.67 | 5.74 | 92.71 |
| | PTQD | 4/8 | 357.66 | **9.24** | 5.42 | 92.46 |
| LDM-4 — PLMS (20 steps) | Ours | 4/8 | **367.22** | 10.26 | **5.11** | **93.44** |
| | Q-Diffusion | 4/4 | N/A | N/A | N/A | N/A |
| | PTQD | 4/4 | N/A | N/A | N/A | N/A |
| | Ours | 4/4 | **185.18** | **10.95** | **11.65** | **75.61** |
| | Q-Diffusion | 2/8 | 56.73 | 34.71 | 12.00 | 49.10 |
| | PTQD | 2/8 | 58.08 | 31.87 | 10.32 | 50.73 |
| | Ours | 2/8 | **182.63** | **6.51** | **6.99** | **79.43** |
| | FP | 32/32 | 373.12 | 11.44 | 6.85 | 93.67 |
| | Q-Diffusion | 8/8 | 365.64 | 10.78 | **6.15** | 92.89 |
| | PTQD | 8/8 | 368.04 | **10.55** | 6.19 | 92.99 |
| | Ours | 8/8 | **370.22** | 11.21 | 6.83 | **93.50** |
| | Q-Diffusion | 4/8 | 351.00 | 9.36 | **6.36** | 91.50 |
| | PTQD | 4/8 | 354.94 | **8.88** | 6.73 | 92.03 |
| LDM-4 — DPM-Solver (20 steps) | Ours | 4/8 | **359.16** | 9.82 | 6.92 | **92.67** |
| | Q-Diffusion | 4/4 | N/A | N/A | N/A | N/A |
| | PTQD | 4/4 | N/A | N/A | N/A | N/A |
| | Ours | 4/4 | **223.40** | **7.54** | **9.47** | **80.06** |
| | Q-Diffusion | 2/8 | 50.12 | 39.08 | 13.75 | 44.67 |
| | PTQD | 2/8 | 51.51 | 38.32 | 12.90 | 46.48 |
| | Ours | 2/8 | **165.76** | **8.55** | **9.71** | **76.76** |

## 5.3 ABLATION STUDY

To assess the efficacy of each proposed component, we conduct a comprehensive ablation study on the ImageNet $256 \times 256$ dataset, employing the LDM-4 model with a DDIM sampler, as presented in Table 3. We initiate the evaluation with a baseline PTQD (He et al., 2023a), which is rooted in the reconstruction-based PTQ method (Li et al., 2021). However, it failed to denoise images when operating under a W4A4 bit-width, yielding an exceedingly high FID score of 259.73. This result underscores the inadequacy of the PTQ method in this particular low bit-width. In sharp contrast, our proposed efficient fine-tuning method QALoRA showcases significant performance improvement with a FID of 11.42, without imposing additional time costs. By incorporating scale-aware LoRA optimization, we achieve an FID reduction of 1.87 and a sFID reduction of 10.95, demonstrating that the LoRA weights are effectively trained to update the quantized weights. By further introducing TALSQ that learns quantization parameters for each denoising step, our method achieves a remarkable sFID of 7.75, putting it in contention with even full-precision models.

Moreover, we conduct a comparative efficiency analysis of PTQ, QAT and our approach across data, GPU memory, and time consumption, as demonstrated in Table 4. The PTQ method (He et al.,

Table 3: The effect of different components proposed in the paper. The experiment is conducted over LDM-4 model on ImageNet $256 \times 256$.

| Method | Bit-width (W/A) | Time Costs (GPU hours) | IS↑ | FID↓ | sFID↓ | Precision↑ (%) |
|---|---|---|---|---|---|---|
| FP | 32/32 | - | 364.73 | 11.28 | 7.71 | 93.66 |
| PTQD | 4/4 | 2.88 | 2.16 | 259.73 | 329.01 | 0.00 |
| QALoRA | 4/4 | 2.60 | 156.15 | 11.42 | 24.18 | 78.36 |
| +scale-aware LoRA optimization | 4/4 | 2.60 | 172.96 | 9.55 | 13.23 | 81.87 |
| +TALSQ | 4/4 | 3.05 | **250.90** | **6.17** | **7.75** | **86.02** |

2023a) emerges as the most efficient in terms of data, time and memory consumption. However, it cannot maintain denoising capabilities at W4A4 precision. On the other hand, QAT method (Park et al., 2022), despite achieving lower FID, necessitates the original training data and incurs a $2.2\times$ increase in memory resources (16950 MB vs. 7538 MB), and a $16.8\times$ increase in time consumption (18.5 GPU hours vs. 1.1 GPU hours) during the quantization of the LDM-8 model on the LSUN-Churches dataset. Comparatively, our proposed approach offers a compelling alternative. It operates without the need for training data and incurs less memory and time consumption compared to the QAT method. Specifically, when quantizing DDIM models on CIFAR-10 dataset, our method attains a FID *that is comparable to that of the QAT method (10.48 vs. 7.30), while enjoying similar time efficiency to the PTQ method (0.97 GPU hours vs. 0.95 GPU hours)*, rendering it a practical choice for real-world applications. Additional ablation experiments can be found in Appendix D.

Table 4: Efficiency comparisons of various quantization methods across training data, GPU memory and time. 'OOM' denotes out-of-memory on RTX3090 GPU. We employ DDIM models for CIFAR-10 dataset and LDM models for other datasets. Both weights and activations of models are quantized to 4-bit.

| Method | Dataset | Training data | GPU Memory (MB) | GPU Time (hours) | FID |
|---|---|---|---|---|---|
| PTQ (He et al., 2023a) | CIFAR-10 | 0 | 4334 | 0.95 | 181.05 |
| | LSUN-Churches | 0 | 7538 | 1.10 | 321.90 |
| | ImageNet $256 \times 256$ | 0 | 11942 | 2.88 | 259.73 |
| QAT (Esser et al., 2019) | CIFAR-10 | 50K | 9974 | 13.89 | 7.30 |
| | LSUN-Churches | 126K | 16950 | 18.50 | 9.08 |
| | ImageNet $256 \times 256$ | 1.2M | OOM | - | - |
| Ours | CIFAR-10 | 0 | 9554 | 0.97 | 10.48 |
| | LSUN-Churches | 0 | 10980 | 1.14 | 14.34 |
| | ImageNet $256 \times 256$ | 0 | 19842 | 3.05 | 6.17 |

# 6 CONCLUSION

In this paper, we have proposed EfficientDM, an efficient data-free fine-tuning framework for low-bit diffusion models. To commence, we have introduced a quantization-aware variant of low-rank adapters, which can be jointly quantized with model weights, enabling both efficient fine-tuning and low-bit inference. To mitigate the reliance on original datasets, we have employed a distilling framework that transfers the noise estimation capabilities of a full-precision model to its quantized counterpart. To address ineffective learning arising from variations in weight quantization parameters across layers, we have introduced scale-aware LoRA optimization, which adaptively adjusts the gradient scales of LoRA weights in different layers. Moreover, we have introduced TALSQ to address the activation distribution variations across steps. Our extensive experiments have demonstrated the superiority of EfficientDM over previous post-training quantized diffusion models. Notably, even when quantizing both weights and activations of LDM-4 to 4-bit on ImageNet $256 \times 256$, where previous methods failed to denoise, EfficientDM exhibited only a marginal $0.05$ increase in sFID. Moreover, our method has demonstrated superior efficiency compared to QAT-based approaches with minor performance gap, making it a practical choice for real-world applications.

**Limitations and future work.** While EfficientDM can attain QAT-level performance using PTQ-level data and time efficiency, it employs a gradient descent algorithm for optimizing QALoRA parameters. Compared to PTQ-based methods, this places higher demands on GPU memory, particularly with diffusion models featuring massive parameters. To address this, memory-efficient optimization methods can be integrated. Moreover, efficient diffusion models for tasks such as video or 3D generation are still under explored.

**Acknowledgement** This work was supported by National Key Research and Development Program of China (2022YFC3602601).

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

# Appendix

## A  EVALUATION OF UNCONDITIONAL GENERATION ON LSUN

### A.1  EVALUATION WITH 100 DENOISING STEPS

In this section, we expand our evaluation of unconditional image generation using LDM models (Rombach et al., 2022a) on high-resolution LSUN-Bedrooms and LSUN-Churches datasets, as illustrated in Table A. The full-precision LDM models are $7.6\times$ larger in size compared to DDIM (Song et al., 2021) models. Therefore most methods focus on PTQ. Under W8A8 quantization, our approach surpasses all previous methods, including the QAT-based LSQ (Esser et al., 2019), demonstrating superior performance in terms of FID. Under W6A6 quantization, existing PTQ-based methods all exhibit significant performance degradation. For instance, on the LSUN-Bedrooms dataset, the state-of-the-art PTQ-based method ADP-DM (Wang et al., 2023) experiences a substantial FID increase of $6.45$, whereas our method reduces FID increase to a mere $0.19$. Under W4A4 quantization, while all other PTQ-based methods fail to generate meaningful images, our approach still achieves a FID of $10.60$ on LSUN-Bedrooms, surpassing the performance of PTQ4DM (Shang et al., 2023) at W6A6 bit-width. Although LSQ, a QAT-based method, attains a lower FID at W4A4 and W6A6 bit-widths, it demands the original training dataset and consumes considerably more time and computing resources. In contrast, our approach achieves a FID that is only 0.07 higher than LSQ at W6A6 bit-width, striking a better balance between performance and efficiency.

Table A: Performance comparisons of unconditional image generation on LSUN datasets. 'N/A' denotes failed image generation.

| Method | Bitwidth (W/A) | Model Size (MB) | IS↑ | FID↓ | Method | Bitwidth (W/A) | Model Size (MB) | IS↑ | FID↓ |
|---|---|---|---|---|---|---|---|---|---|
| \multicolumn LSUN-Bedrooms LDM-4 (steps = 100) | | | | | \multicolumn LSUN-Churches LDM-8 (steps = 100) | | | | |
| FP | 32/32 | 1045.4 | 2.29 | 3.43 | FP | 32/32 | 1125.2 | 2.70 | 4.08 |
| PTQ4DM | 8/8 | 261.67 | 2.21 | 4.75 | PTQ4DM | 8/8 | 281.85 | 2.52 | 5.54 |
| Q-Diffusion | 8/8 | 261.67 | 2.19 | 4.67 | Q-Diffusion | 8/8 | 281.85 | 2.53 | 4.87 |
| ADP-DM | 8/8 | 261.67 | **2.35** | 3.88 | ADP-DM | 8/8 | 281.85 | 2.69 | 4.02 |
| LSQ | 8/8 | 261.67 | 2.18 | 3.23 | LSQ | 8/8 | 281.85 | 2.68 | 4.06 |
| Ours | 8/8 | 261.69 | 2.27 | **2.98** | Ours | 8/8 | 281.89 | **2.71** | **4.01** |
| PTQ4DM | 6/6 | 196.33 | 2.08 | 11.10 | PTQ4DM | 6/6 | 211.53 | 2.46 | 11.05 |
| Q-Diffusion | 6/6 | 196.33 | 2.11 | 10.10 | Q-Diffusion | 6/6 | 211.53 | 2.47 | 10.90 |
| ADP-DM | 6/6 | 196.33 | 2.27 | 9.88 | ADP-DM | 6/6 | 211.53 | 2.67 | 6.90 |
| LSQ | 6/6 | 196.33 | 2.16 | **3.55** | LSQ | 6/6 | 211.53 | 2.66 | **5.04** |
| Ours | 6/6 | 196.36 | **2.28** | 3.62 | Ours | 6/6 | 211.57 | **2.82** | 6.29 |
| PTQ4DM | 4/4 | 130.99 | N/A | N/A | PTQ4DM | 4/4 | 141.20 | N/A | N/A |
| Q-Diffusion | 4/4 | 130.99 | N/A | N/A | Q-Diffusion | 4/4 | 141.20 | N/A | N/A |
| ADP-DM | 4/4 | 130.99 | N/A | N/A | ADP-DM | 4/4 | 141.20 | N/A | N/A |
| LSQ | 4/4 | 130.99 | 2.11 | **6.17** | LSQ | 4/4 | 141.20 | 2.63 | **9.08** |
| Ours | 4/4 | 131.02 | **2.27** | 10.60 | Ours | 4/4 | 141.24 | **2.81** | 14.34 |

### A.2  EVALUATION WITH 20 DENOISING STEPS

In this section, we perform experiments on LSUN-Bedrooms with 20 steps to validate the effectiveness of interpolated quantization scales, as shown in Table B. Notably, our approach introduces only a slight FID increase of $0.01$ and $0.64$ for W8A8 and W6A6 bit-widths, respectively, which demonstrates the effectiveness the interpolated quantization scales in the fewer-step regime.

## B  DEPLOYMENT EFFICIENCY

In this section, we evaluated the latency of matrix multiplication and convolution operations in both quantized and full-precision diffusion models, utilizing an RTX3090 GPU and the CUTLASS (Kerr et al., 2017) implementation, as demonstrated in Table C. When both weights and activations are quantized to 8-bit, we observe a $3.74\times$ reduction in model size and a $2.03\times$ reduction in latency compared to their full-precision counterparts. Furthermore, with weights and activations quantized

Table B: Performance of unconditional image generation on LSUN-Bedrooms with 20 steps. 'N/A' denotes failed image generation.

| Method | Bitwidth (W/A) | Model Size (MB) | IS↑ | FID↓ |
|---|---|---|---|---|
| FP | 32/32 | 1045.4 | 2.21 | 9.74 |
| Q-Diffusion | 8/8 | 261.67 | 2.13 | 10.10 |
| Ours | 8/8 | 261.69 | **2.20** | **9.75** |
| Q-Diffusion | 6/6 | 196.33 | **3.78** | 149.39 |
| Ours | 6/6 | 196.36 | 2.27 | **10.39** |
| Q-Diffusion | 4/4 | 130.99 | N/A | N/A |
| Ours | 4/4 | 131.02 | **2.24** | **16.35** |

to 4-bit, we observe a reduction in model size by $6.58\times$ and an increased speedup of $3.34\times$. In the case of W2A8 quantization, the model size is further reduced by $10.57\times$. However, there is currently no hardware support for W2A8 precision on Nvidia's GPUs. Therefore, the calculations are still conducted under W8A8 precision.

Table C: Comparisons of time cost across various bitwidth configurations on ImageNet $256 \times 256$.

| Model | Bitwidth (W/A) | Model Size (MB) | IS↑ | FID↓ | sFID↓ | Precision↑ (%) | Time (ms) |
|---|---|---|---|---|---|---|---|
| | 32/32 | 1603.68 | 364.73 | 11.28 | 7.70 | 93.66 | 436.8 |
| LDM-4 — DDIM (20 steps) | 8/8 | 427.65 | 362.34 | 11.38 | 8.04 | 93.77 | 214.4 |
| | 4/4 | 243.61 | 250.90 | 6.17 | 7.75 | 86.02 | 130.4 |
| | 2/8 | 151.60 | 175.03 | 7.60 | 8.12 | 78.90 | 214.4 |

## C  VARIATION OF ACTIVATION DISTRIBUTIONS ACROSS STEPS

In this section, we present the value ranges of activations across various time steps in the LDM-4 model pretrained on ImageNet $256 \times 256$ dataset, as illustrated in Figure A. Previous PTQ methods for diffusion models employ a single set of quantization parameters for activations across all time steps. However, this approach can result in significant quantization errors when dealing with steps with distinct activation distributions (such as step 0 in Figure A), thereby leading to suboptimal denoising performance. In contrast, our proposed TALSQ addresses this issue by assigning different quantization parameters to individual time steps, a scheme that can be optimized during the fine-tuning process. We further visualize the learned quantization scales and the variance of the activations in Figures B and C. Layers exhibiting significant shifts in activation distribution show considerable variation in the learned scales across different steps (see Figure Ba), and vice versa in the remaining subfigures.

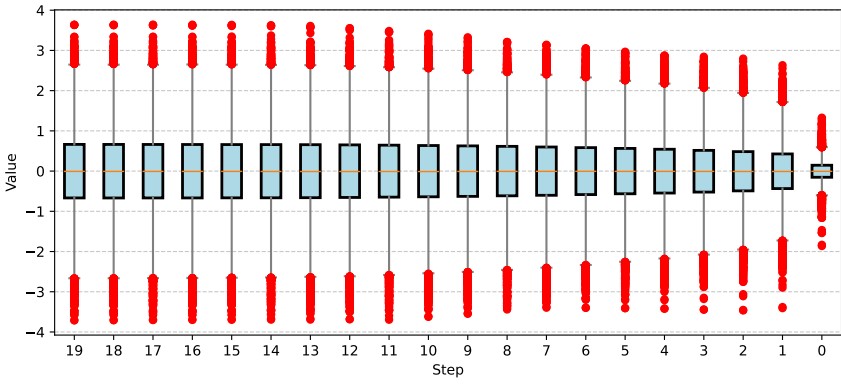

Figure A: Ranges of model output across various steps. Results are obtained by LDM-4 model on ImageNet $256 \times 256$ dataset.

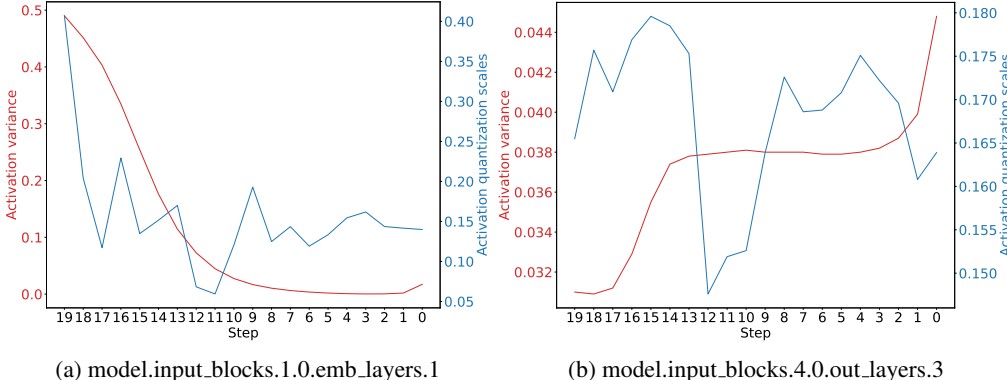

(a) model.input_blocks.1.0.emb_layers.1    (b) model.input_blocks.4.0.out_layers.3

Figure B: The variance of activations and the learned quantization scales of intermediate layers across different steps. Caption denotes the layer name. Data is collected by W4A4 LDM-4 model on ImageNet $256 \times 256$.

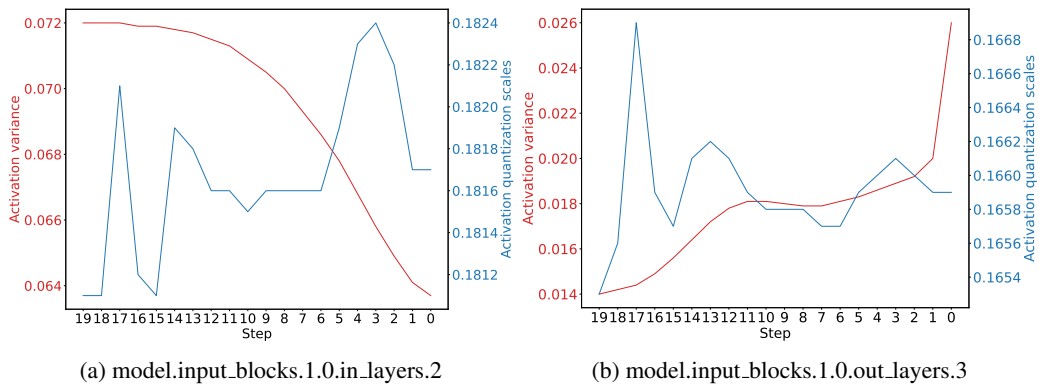

(a) model.input_blocks.1.0.in_layers.2    (b) model.input_blocks.1.0.out_layers.3

Figure C: The variance of activations and the learned quantization scales of intermediate layers across different steps. Caption denotes the layer name. Data is collected by W4A4 LDM-4 model on LSUN-Bedrooms.

# D    ADDITIONAL ABLATION STUDY

## D.1    ABLATION STUDY OF TALSQ

In this section, we compare the performance of TALSQ with dynamic quantization and time-step-wise post-training quantization. The experiments are conducted over ImageNet $256 \times 256$ with W4A4 LDM-4 models. As shown in Table D, the naive Min-Max dynamic quantization fails to denoise images, even with QALoRA weights fine-tuned. Time-step-wise quantization, obtained through post-training techniques with calibration sets, outperform non-time-step-wise quantization by incorporating temporal information. Utilizing TALSQ further improves performance, resulting in a reduction of $1.64$ in FID and $2.03$ in sFID, showcasing the superiority of the learned temporal quantization scales.

## D.2    SELECTION OF RANK $r$

In this section, we conduct ablation experiments to explore the impact of the rank parameter $(r)$ in LoRA on its performance, as outlined in Table E. When the rank is set below 32, the model experiences a substantial decrease in performance following the fine-tuning process. However, when the rank is set at 32, $2.39\%$ of model parameters are trainable, resulting in a $4.61$ decrease in FID. Increasing the rank further to 64 leads to a $2.27\%$ increase in trainable parameters, with only a marginal $0.09$ reduction in FID. Consequently, we set $r = 32$ as the default value for all experiments.

Table D: Performance comparisons of various time-step-wise quantization scheme on ImageNet $256 \times 256$.

| Method | Bit-width (W/A) | IS↑ | FID↓ | sFID↓ | Precision↑ |
|---|---|---|---|---|---|
| baseline (QALoRA+scale-aware LoRA optimization) | 4/4 | 172.96 | 9.55 | 13.23 | 81.87 |
| baseline + dynamic quantization | 4/4 | N/A | N/A | N/A | N/A |
| baseline + PTQ time-step-wise quantization | 4/4 | 224.02 | 7.81 | 9.78 | 82.42 |
| baseline + TALSQ | 4/4 | **250.90** | **6.17** | **7.75** | **86.02** |

Table E: Performance comparisons under different rank of LoRA. Data is collected over 6-bit LDM-8 model on LSUN-Churches.

| | $r=0$ | $r=8$ | $r=16$ | $r=32$ | $r=64$ |
|---|---|---|---|---|---|
| FID↓ | 10.90 | 10.53 | 8.06 | 6.29 | 6.20 |
| Trainable Parameters (%) | 0 | 0.61 | 1.20 | 2.39 | 4.66 |

## E  GRADIENT ANALYSIS OF QALoRA

In this section, we further analyze the gradient of QALoRA, elucidating the reasons behind its ineffective learning. Referring to Eq. (6) and considering the impact of STE (Bengio et al., 2013), the gradient of **BA** can be expressed as:

$$\frac{\partial \mathbf{Y}}{\partial(\mathbf{BA})} = \frac{\partial \mathbf{Y}}{\partial \hat{\mathbf{W}}} \frac{\partial \hat{\mathbf{W}}}{\partial(\mathbf{BA})} = \hat{\mathbf{X}}^\top. \tag{A}$$

The key insight is that the gradient magnitudes of LoRA weights remain unaffected by the quantization scale, which is exactly the step size of the quantization step-like function. Consequently, in layers with larger quantization scales, the minor updates to LoRA weights will be diminished by the round operation.

## F  ADDITIONAL VISUALIZATION RESULTS

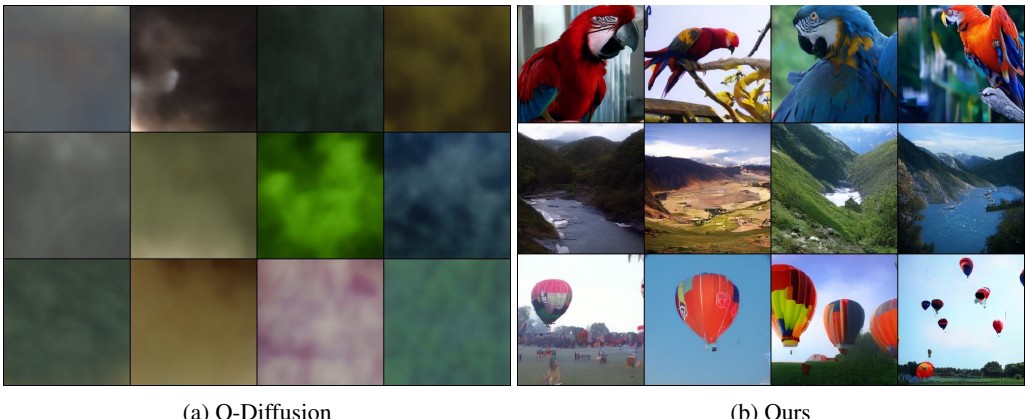

(a) Q-Diffusion                    (b) Ours

Figure D: Samples generated by W4A4 LDM model on ImageNet $256 \times 256$.

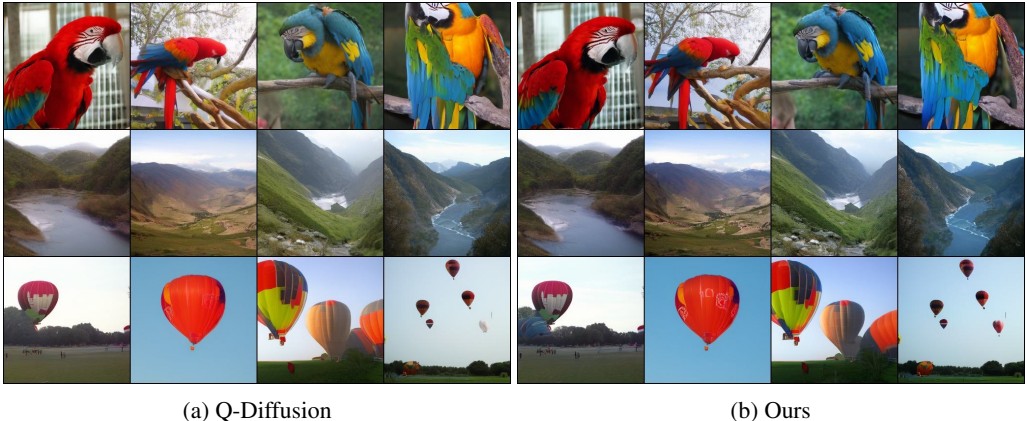

|(a) Q-Diffusion|(b) Ours|

Figure E: Randomly generated samples by W4A8 LDM model on ImageNet $256 \times 256$.

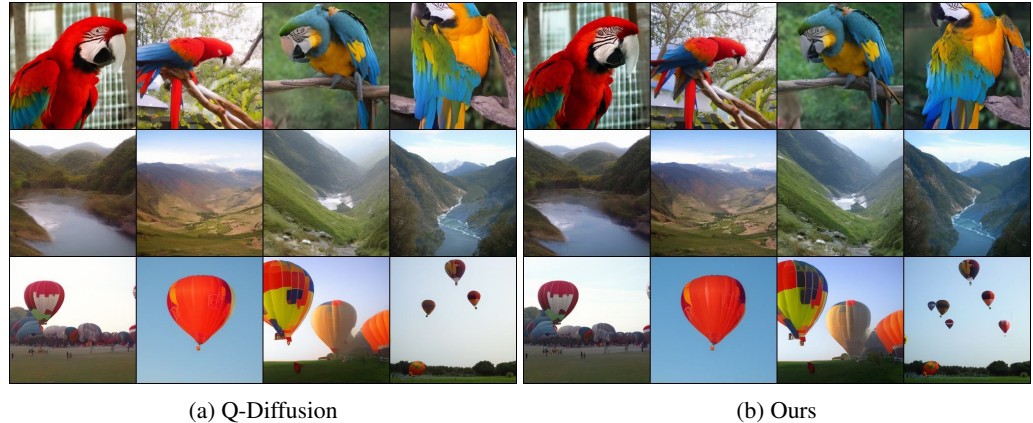

|(a) Q-Diffusion|(b) Ours|

Figure F: Randomly generated samples by W8A8 LDM model on ImageNet $256 \times 256$.

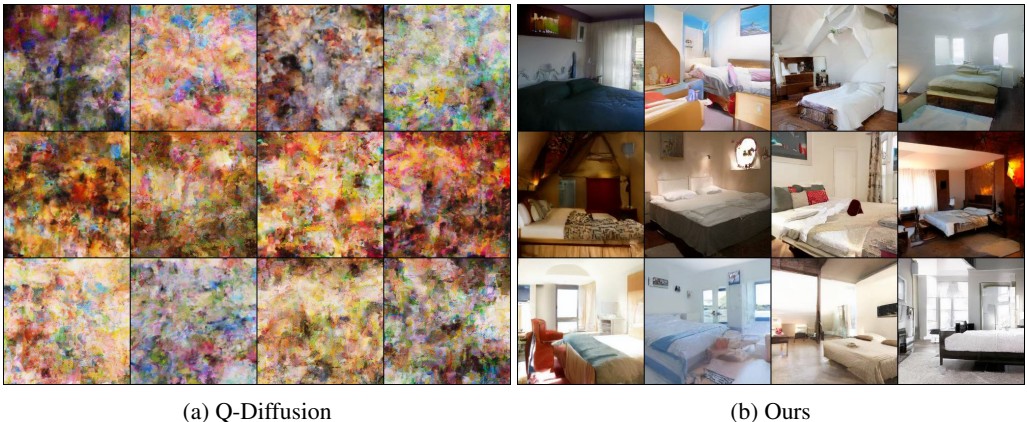

|(a) Q-Diffusion|(b) Ours|

Figure G: Randomly generated samples by W4A4 LDM model on LSUN-Bedrooms.

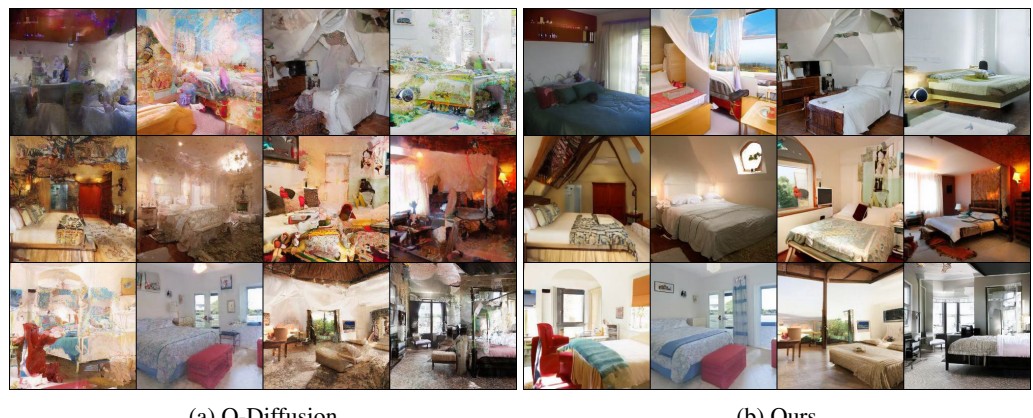

(a) Q-Diffusion                                    (b) Ours

Figure H: Randomly generated samples by W6A6 LDM model on LSUN-Bedrooms.

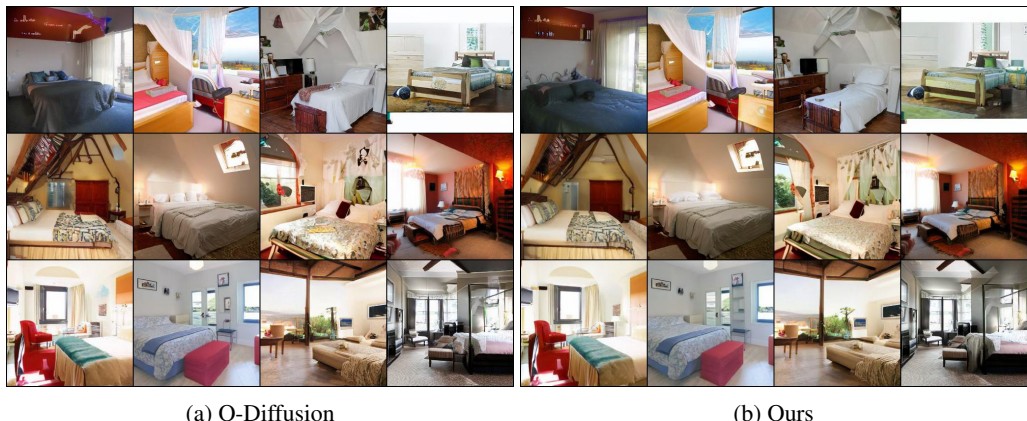

(a) Q-Diffusion                                    (b) Ours

Figure I: Randomly generated samples by W8A8 LDM model on LSUN-Bedrooms.

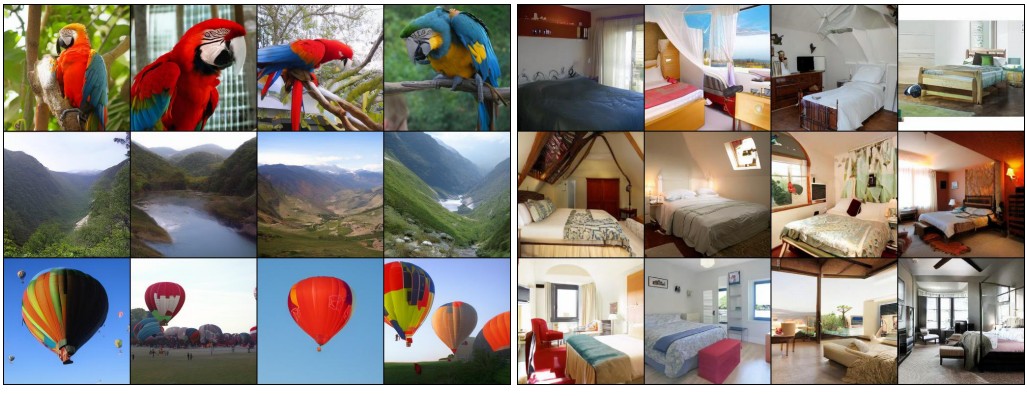

(a) ImageNet $256 \times 256$ samples

(b) LSUN-Bedrooms samples

Figure J: Samples generated by full-precision LDM models.

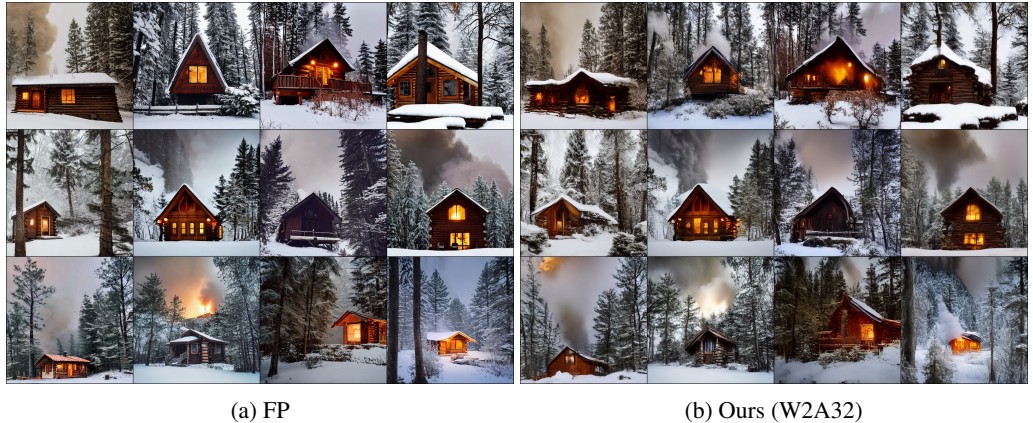

(a) FP

(b) Ours (W2A32)

Figure K: Randomly generated samples by FP and W2A32 Stable Diffusion model with prompt "*A cozy cabin nestled in a snowy forest with smoke rising from the chimney*".

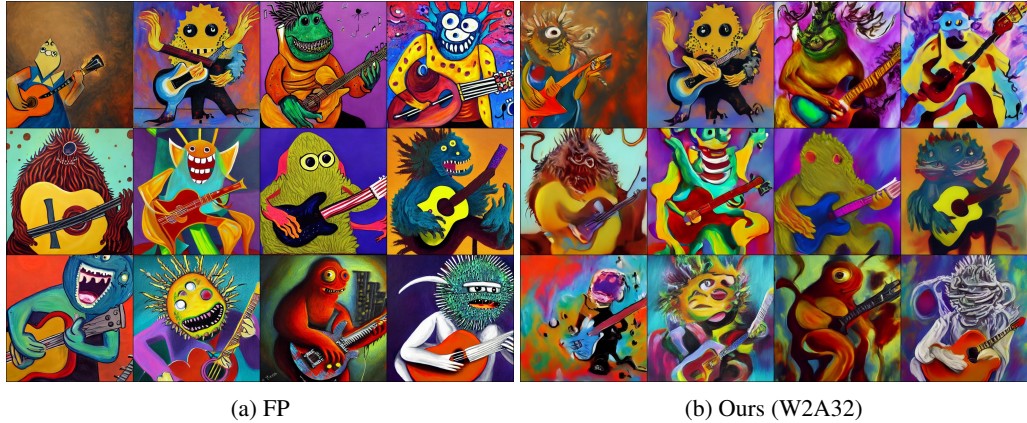

(a) FP

(b) Ours (W2A32)

Figure L: Randomly generated samples by FP and W2A32 Stable Diffusion model with prompt "*a painting of a virus monster playing guitar*".

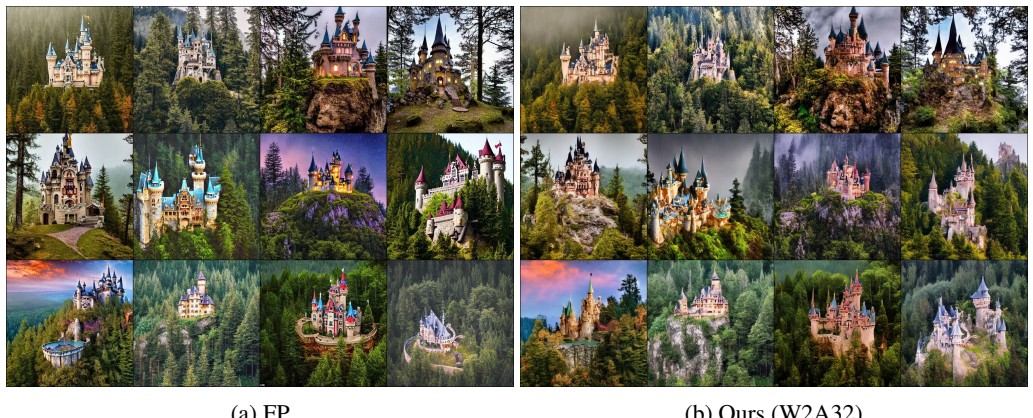

(a) FP                                    (b) Ours (W2A32)

Figure M: Randomly generated samples by FP and W2A32 Stable Diffusion model with prompt "*a magical fairy tale castle on a hilltop surrounded by a mystical forest*".

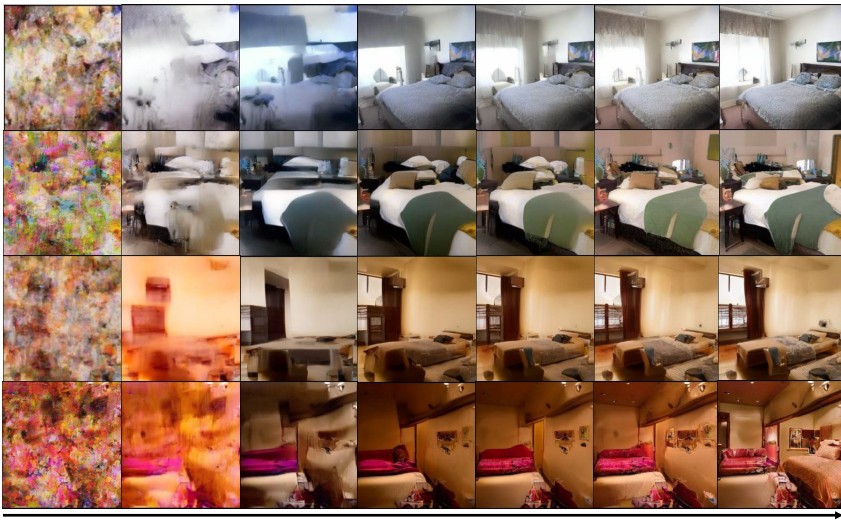

EfficientDM Finetuning

Figure N: Visualization of samples generated by PTQD (the leftmost column) and EfficientDM during fine-tuning process. Results are obtained by 4-bit LDM model on LSUN-Bedrooms dataset. PTQD fails to denoise images when both weights and activations are quantized to 4-bit, while EfficientDM recovers its strong capability to generate high-quality images through the proposed efficient fine-tuning framework.

