# OpenReview forum: "EfficientDM: Efficient Quantization-Aware Fine-Tuning of Low-Bit Diffusion Models"
_ICLR.cc/2024/Conference — ICLR 2024 spotlight_

### Official Review · Reviewer_SYoS · 2023-11-01

**Soundness:** 3 good
**Presentation:** 3 good
**Contribution:** 3 good
**Rating:** 6
**Confidence:** 3

**Summary:**

The paper proposes a new quantization scheme for diffusion models. In particular, the paper notes that post-training quantization (PTQ) may be efficient but brings relatively low performance compared to quantization-aware training (QAT). Conversely, QAT brings higher performance but requires heavy computational resources. To combine the advantages of these two main quantization approaches, the paper introduces a quantization counterpart of low-rank adapter (LoRA). The paper also proposes to quantize the model in a data-free manner through distillation from the original full-precision model.

**Strengths:**

- The paper introduces a new quantization method that brings substantial efficiency improvement without incurring extra computational overhead and performance degradation.

- The paper demonstrates strong performance.

- The paper is clearly written.

**Weaknesses:**

- Comparisons: What happens if other quantization models also employ LoRA and distillation, which are common techniques to use?

- Novelty concern: I think the paper presents a combination of existing works (scale-aware optimization from LSQ, common distillation technique, and LoRA). Can the authors clarify the difference in contribution from the combination of existing works? If there is a difference, how does the performance differ compared to the combination?

**Questions:**

Please refer to the weaknesses section.

---

> ### Author Response · Authors · 2023-11-17
> **Rebuttal by Authors**
>
> Thanks to the reviewer for the valuable comments.
>
> **Q1: Clarify the difference in contribution from the combination of existing works (scale-aware optimization from LSQ, common distillation technique, and LoRA).**
> In this paper, we propose a data-free, quantization-aware and parameter-efficient fine-tuning framework for low-bit diffusion models, which seamlessly integrates three tightly coupled components, QALoRA, scale-aware optimization and data-free distillation.
> Overall, our work has been well recognized by other reviewers, such as  "the idea of QALoRA is novel" (Reviewer 5Wot), "this paper introduces low rank adapter and distillation loss ...is the first to achieve very good performance..." (Reviewers ec6g) and "the proposed QALoRA is data-free and efficient." (Reviewer BLfX).
>
> To elaborate, **our scale-aware optimization differs from the ``STEP SIZE GRADIENT SCALE" used in LSQ in both formula and function.** From a formula standpoint, we adjust the gradient of QALoRA weight using the averaged quantization scale, whereas LSQ adjusts the gradient of quantization scales using the number of weights and precision. The goal of our technique is to improve the learning effectiveness of QALoRA, a critical issue in its optimization. As referred to Table 3 in the paper, our scale-aware optimization significantly improves EfficientDM's performance, leading to a 10.95 decrease in sFID.
>
> Common distillation methods require access to the training dataset, which may be challenging due to massive size or copyright concerns. The recent work LLM-QAT [ii] proposes a distillation method that utilizes generations produced by full-precision (FP) models. However, it requires generating massive amounts of data using the teacher model beforehand and training the entire quantized model, incurring significant training costs. **Our distillation scheme stands out by not requiring the generation and storage of any data.** Instead, we input the same random Gaussian noise to both FP and quantized models and minimize the mean squared error between their denoising results. The distillation process is conducted in a data-free manner, which is efficient and easy to implement.
>
> **The proposed QALoRA is also distinct from the existing work QLoRA [iii], which applies LoRA to quantized models.** As referred to ``Quantization-aware low-rank adapter" part in the paper, QLoRA incurs limitations when both weights and activations are quantized. The full-precision LoRA weights can not be merged with quantized model weights, leading to increased model size and massive floating-point calculations during inference. In contrast, the proposed QALoRA is a new variant of LoRA that can be seamlessly integrated with quantized model weights. After fine-tuning, there is no additional storage needed and the calculations between quantized weights and activations can be efficiently implemented for faster inference.
>
> **Q2: What happens if other quantization models also employ LoRA and distillation, which are common techniques to use?** As referred to Q1, both QALoRA and our distillation scheme are distinct from previous methods and they are seamlessly integrated within the proposed data-free, quantization-aware fine-tuning framework for low-bit diffusion models.
>
> [i] Esser, Steven K., et al. ``Learned step size quantization." ICLR 2020.
>
> [ii] Liu, Zechun, et al. ``LLM-QAT: Data-Free Quantization Aware Training for Large Language Models." arXiv 2023.
>
> [iii] Dettmers, Tim, et al. "Qlora: Efficient finetuning of quantized llms." NeurIPS 2023.

---

> ### Comment · Reviewer_SYoS · 2023-11-21
>
> I appreciate the authors' response.
> I acknowledge the differences in the distillation technique.
> However, a part of my concerns remain regarding the novelty.
> In the main text, it says "Inspired by LSQ (Esser et al., 2019), a technique where quantization scales are optimized alongside other trainable parameters through the gradient descent algorithm, we extend the activation quantization scale into the temporal domain."
> According to the text, it seems like the proposed scale-aware quantization is a simple extension of the one proposed in LSQ.
> I do not think simple extention of existing work to temporal domain is novel enough to be considered as one of main contributions.
> Please clarify if there is any misunderstanding.

---

> ### Author Response · Authors · 2023-11-21
> **Response to Reviewer SYoS**
>
> Dear Reviewer SYoS,
>
> Thank you for your feedback. It appears there may be some misunderstandings regarding our proposed techniques. The scale-aware optimization and Temporal LSQ (TLSQ) are distinct techniques proposed in our paper. As referred to ``Variation of weight quantization scales across layers" part of the paper, **the scale-aware optimization aims to improve the learning effectiveness of QALoRA due to variations in weight quantization scales across different layers, which is not an extension of LSQ.**
>
> In addressing the temporal dynamics of activation distribution across steps, we introduce Temporal LSQ (TLSQ), which is outlined in the "Variation of activation distribution across steps" part of the paper. To contextualize our innovation, we highlight a comparison with a related approach TDQ [i], as referred to Q1 in the general response. Unlike TDQ, which employs additional MLP modules to approximate quantization scales, TLSQ optimizes temporal quantization scales directly, offering a more computationally efficient approach. As referred to Tables 1 and 3 in the paper, TLSQ greatly improves the performance of EfficientDM, **outperforming TDQ in both performance and efficiency** under W8A8 and W4A8 bit-widths on CIFAR-10 dataset. **These results affirm the advantages and contributions of the proposed TLSQ technique.**
>
> If there are any additional concerns you have, please feel free to let us know.
>
> Thank you once again for your time, guidance, and consideration.
>
> Best regards,
>
> Authors of #3007.
>
> [i] So J, Lee J, Ahn D, et al. "Temporal Dynamic Quantization for Diffusion Models." NeurIPS, 2023.

---

> > ### Comment · Reviewer_SYoS · 2023-11-21
> >
> > Thanks for the clarification.
> > Then, it seems the text I'm refering to is a bit misleading.
> > It would be great if the text is revised to avoid misunderstanding.
> > Other than that, my concerns have been addressed.

---

> > > ### Author Response · Authors · 2023-11-21
> > > **Response to Reviewer SYoS**
> > >
> > > Dear Reviewer SYoS,
> > >
> > > Thank you for your great efforts and constructive feedback in reviewing our paper. We have revised the text you referred to in order to eliminate potential misunderstandings.
> > >
> > > Once again, thank you for your time and commitment in reviewing our work.
> > >
> > > Best regards,
> > >
> > > Authors of #3007.

---

### Official Review · Reviewer_BLfX · 2023-11-07

**Soundness:** 3 good
**Presentation:** 3 good
**Contribution:** 2 fair
**Rating:** 6
**Confidence:** 4

**Summary:**

This paper presents a data-free and efficient Quantization-Aware Training (QAT) method for diffusion models. For efficient QAT, it introduces the Quantization-Aware Low-Rank Adapter (QALoRA), which combines LoRA and QAT. The paper extends the LSQ QAT method a little to the Temporal LSQ method, which learns different scale factors for different time steps to handle the activation distribution difference across steps. The experimental of image diffusion and latent diffusion models on CIFAR-10, LSUN, and ImageNet demonstrates that this method can significantly outperform previous PTQ methods when doing W4A4 and W2 quantization.

**Strengths:**

- Applying QAT to diffusion models to achieve better quantization performance is reasonable.
- The proposed QALoRA is data-free and efficient (cost about 10 GPU hours).
- The experimental results are promising.
- Actual speedup with CUTLASS is reported.

**Weaknesses:**

There exist some formulas and details that are not clear enough. Some additional ablation and analysis experiments are needed to make the overall method more convincing. Check the questions section.

**Questions:**

- How to get $\bf{x}_t$ in equation (7) is not described properly. Do the authors sample $\bf{x}_T \sim \mathcal{N}$, and run several solver steps using the FP model to get $\bf{x}_t$, or otherwise? If it is the case, the equation (7) is not written properly.
- The proposed Temporal LSQ (TLSQ) method uses a different activation quantization scale for different time steps. Can the authors show the learned scales and analyze how the scale factors of certain layers change w.r.t. the time steps on different datasets?
- Can the authors compare TLSQ with deciding the time-step-wise activation quantization scale using some calibration data or even run-time dynamic quantization? This can help illustrate the necessity of LSQ.
- The paper mentioned that "we interpolate the learned temporal quantization scales to deal with the gap of sampling steps between fine-tuning and inference". I found steps=100 experiments on CIFAR-10 and LSUN in Table 1 and Appendix A, I wonder if the authors experimented with using a different schedule with fewer steps? Does this scale-deciding technique work well in the fewer-step regime?
- Is QALoRA applied for all the weights, including the convolutions and the attention layers?
- For the good of future efficient diffusion, can the authors discuss more relevant limitations and raise questions worth future studying? The current discussion is not specific.

---

> ### Author Response · Authors · 2023-11-17
> **Rebuttal by Authors**
>
> Thanks to the reviewer for the valuable comments.
>
> **Q1: How to get $\mathbf{x}_t$ in Eq. (7)? The Eq. (7) is not written properly.** Please refer to Q2 in general response. The input data $\mathbf{x}_t$ is derived from denoising random Gaussian noise $\mathbf{x}_T \sim \mathcal{N}(0,1)$ with the FP model through $T-t$ iterations. We have revised the explanation of Eq. (7) in the revised paper.
>
> **Q2: Can the authors show the learned scales and analyze how the scale factors of certain layers change on different datasets?** As referred to Figure B to C in the supplementary material of revised paper, we visualize the learned quantization scales and the variance of the activations collected from W4A4 LDM-4 model over ImageNet $256\times256$ and LSUN-Bedrooms. Layers exhibiting significant shifts in activation distribution show considerable variation in the learned scale across different steps, and vice versa.
>
> **Q3: Can the authors compare TLSQ with deciding the time-step-wise activation quantization scale using some calibration data or dynamic quantization?** We conduct ablation experiments to compare the performance of TLSQ with dynamic quantization and time-step-wise post-training quantization, as referred to Table D in the supplementary material of revised paper. The naive Min-Max dynamic quantization fails to denoise images, even with QALoRA weights fine-tuned. Time-step-wise quantization scales, obtained through post-training techniques with calibration sets, outperform non-time-step-wise quantization by incorporating temporal information. Utilizing TLSQ further improves performance, resulting in a reduction of $1.64$ in FID and $2.03$ in sFID, showcasing the superiority of learned temporal quantization scales.
>
> | Method                                                                                | Bit-width (W/A) | IS$\uparrow$ | FID$\downarrow$ | sFID$\downarrow$ | Precision$\uparrow$ |
> | :---: | :---: | :---: | :---: | :---: | :---: |
> | baseline (QALoRA+scale-aware optimization)                                            | 4/4         | 172.96     | 9.55          | 13.23          | 81.87              |
> | baseline + dynamic quantization                                                      | 4/4         | N/A        | N/A           | N/A            | N/A                |
> | baseline + PTQ time-step-wise quantization                                         | 4/4         | 224.02     | 7.81          | 9.78           | 82.42              |
> | baseline + TLSQ                                                                     | 4/4         | **250.90**  | **6.17**      | **7.75**       | **86.02**          |
>
>
>
> **Q4: Does scale-deciding technique work well in the fewer-step regime?** As referred to Table 2 in the paper, the performance on ImageNet $256\times256$ is evaluated with 20 steps, which demonstrates the strong performance of EfficientDM in the fewer-step regime. Moreover, we conduct experiments on LSUN dataset with 20 steps, as referred to Table B in the supplementary material of revised PDF. The proposed EfficientDM consistently outperforms previous quantized diffusion models in the fewer-step regime, which demonstrates the effectiveness of the interpolated quantization scales.
>
> **Performance of unconditional image generation on LSUN-Bedrooms with 20 steps. `N/A' denotes failed image generation.**
> | Method | Bitwidth (W/A) | Model Size (MB) | IS$\uparrow$ | FID$\downarrow$ |
> | :---: | :---: | :---: | :---: | :---: |
> | FP     | 32/32       | 1045.4      | 2.21        | 9.74          |
> | Q-Diffusion | 8/8        | 261.67      | 2.13        | 10.10         |
> | Ours   | 8/8        | 261.69      | **2.20**    | **9.75**      |
> | Q-Diffusion | 6/6        | 196.33      | **3.78**     | 149.39        |
> | Ours   | 6/6        | 196.36      | 2.27        | **10.39**     |
> | Q-Diffusion | 4/4        | 130.99      | N/A         | N/A           |
> | Ours   | 4/4        | 131.02      | **2.24**    | **16.35**     |
>
> **Q5: Is QALoRA applied for all the weights, including the convolutions and the attention layers?** As referred to ``Quantization settings" in Section 5.1 in the paper, the input embedding and output layers are fixed to 8-bit, whereas other convolutional and fully-connected layers are applied with QALoRA modules.
>
> **Q6: Discuss limitations and raise questions worth future studying.** Thanks for your valuable comment. We have updated the ``Limitations and future work" part in the revised paper. Specifically, while EfficientDM can attain QAT-level performance using PTQ-level data and time efficiency, it employs a gradient descent algorithm for optimizing QALoRA parameters. Compared to PTQ-based methods, this places higher demands on GPU memory, particularly with diffusion models featuring massive parameters. To address this, memory-efficient optimization methods can be integrated. Moreover, efficient diffusion models for tasks such as video or 3D generation are still under explored.

---

> > ### Comment · Reviewer_BLfX · 2023-11-23
> > **Thanks for the response**
> >
> > Thanks for the authors' response. It addressed most of my questions. I keep my original rating.
> >
> > How about a step number that is fewer than 20?

---

> > > ### Author Response · Authors · 2023-11-23
> > > **Appreciation for Your Valuable Feedback**
> > >
> > > Dear Reviewer BLfX,
> > >
> > > Thank you for your feedback. We truly appreciate your careful consideration of our responses. Since the discussion phase is closing very soon, we will include the results of a few-step sampling schedule in the final version.
> > >
> > > Best regards,
> > >
> > > Authors of #3007.

---

### Official Review · Reviewer_5Wot · 2023-11-09

**Soundness:** 4 excellent
**Presentation:** 3 good
**Contribution:** 4 excellent
**Rating:** 8
**Confidence:** 5

**Summary:**

This paper introduces a data-free fine-tuning framework tailored for low-bit diffusion models. The key approach involves freezing the pretrained diffusion model and fine-tuning a set of quantization-aware LoRA variants (QALoRA) by employing knowledge distillation to capture the denoising capabilities in the full-precision model. The paper also introduces two techniques, namely scale-aware optimization and learned step-size quantization, to address challenges related to ineffective learning of QALoRA and variations in activation distributions. Extensive experiments highlight that EfficientDM achieves performance levels comparable to QAT methods while preserving the data and time efficiency advantages of PTQ methods.

**Strengths:**

1.	Achieving QAT-level performance with PTQ-level efficiency is significant and promising for low-bit diffusion models.
2.	The idea of the QALoRA is novel. Compared to QLoRA, it avoids extra floating-point calculations during inference.
3.	The results are encouraging and demonstrate the strong performance of EfficientDM under various bit-widths.
4.	The paper is well-organized and easy to follow. The supplementary material provides additional experimental results and comprehensive visualization results, which enhance the overall credibility of the work.

**Weaknesses:**

1.	It would be beneficial to evaluate EfficientDM over recent text-to-image diffusion models, such as Stable Diffusion.
2.	Recent work TDQ [1] also introduces a quantization method that adjusts the quantization scale at various denoising steps. The differences should be discussed.
3.	Formulating the gradient of LoRA weights can help elucidate the reasons for ineffective learning of QALoRA.
4.	Figure 2: the notation for scale-aware optimization is inconsistent with Eq. (8), please fix it.

[1] So, Junhyuk, et al. “Temporal Dynamic Quantization for Diffusion Models.” arxiv 2023.

**Questions:**

See weaknesses

---

> ### Author Response · Authors · 2023-11-17
> **Rebuttal by Authors**
>
> Thanks to the reviewer for the valuable comments.
>
> **Q1: Evaluate EfficientDM over recent text-to-image diffusion models.**
> As referred to Figures K-M in the supplementary material, we include visualization results to evaluate EfficientDM over text-to-image Stable Diffusion models.
>
> **Q2: The differences from TDQ should be discussed.** Please refer to Q1 in the general response. TDQ incorporates additional MLP modules for each quantized layer to approximate quantization scales. This approach entails training these extra MLP modules, leading to a significant surge in training expenses. Conversely, TLSQ introduces a limited number of adjustable quantization scales per layer, making TLSQ more computationally efficient than TDQ. Empirically, TLSQ achieves a 20.9% faster training iteration speed compared to TDQ over DDIM model on CIFAR-10 dataset.
>
> **Q3: Formulating the gradient of LoRA weights.**  Referring to Eq.~(6) in the paper and considering the impact of STE [i], the gradient of LoRA weights $\mathbf{BA}$ can be expressed as:
> \begin{gather}
>     \frac{\partial \mathbf{Y}}{\partial (\mathbf{BA})}= \frac{\partial \mathbf{Y}}{\partial \hat{\mathbf{W}}} \frac{\partial \hat{\mathbf{W}}}{\partial (\mathbf{BA})} = \hat{\mathbf{X}}^\top
> \end{gather}
> The key insight is that the gradient magnitudes of LoRA weights remain unaffected by the quantization scale, which is exactly the step size of the quantization step-like function. Consequently, in layers with larger quantization scales, the minor updates to LoRA weights will be diminished by the round operation, which causes the ineffective learning of QALoRA.
>
> **Q4: The notation for scale-aware optimization in Fig. (2) is inconsistent with Eq. (8).** Thanks for the valuable comments. We have corrected it in the revised version.
>
> [i] Bengio, Yoshua, Nicholas Léonard, and Aaron Courville. "Estimating or propagating gradients through stochastic neurons for conditional computation." arXiv, 2013.

---

> > ### Comment · Reviewer_5Wot · 2023-11-23
> >
> > Thanks for the authors' reply, and my questions are addressed very well. Overall, I like this paper since the motivation is very clear, and the proposed method is novel and brings new insights. I keep my score and recommend this paper for acceptance.

---

> > > ### Author Response · Authors · 2023-11-23
> > > **Appreciation for Your Valuable Feedback**
> > >
> > > Dear Reviewer 5Wot,
> > >
> > > Thank you for your feedback. We truly appreciate your careful consideration of our responses.
> > >
> > > Best regards,
> > >
> > > Authors of #3007.

---

### Official Review · Reviewer_ec6g · 2023-11-09

**Soundness:** 3 good
**Presentation:** 3 good
**Contribution:** 2 fair
**Rating:** 6
**Confidence:** 5

**Summary:**

This paper proposes a quantization-aware variant of low rank adapter and a data-free training scheme  for fine-tuning quantized diffusion models. It introduces scale-aware techniques to optimize the weight quantization parameters. For activation quantization, this paper employs a separate activation quantization step-size parameter for each denoising time step. With tuning the low rank weight parameter adapters, this method can achieve  image generation performance comparable to QAT based methods with much lower fine-tuning cost. It firstly achieves FID score as low as 6.17 on conditional image generation on ImageNet 256x256 dataset with 4bit-weight, 4bit-activation diffusion model.

**Strengths:**

* This paper is the first to achieve very good image generation performance with W4A4 diffusion models and W2A8 diffusion models.

* This paper introduces **low rank adapter** and **distillation loss** to fine-tune quantized diffusion models and achieve good results with relatively low cost than QAT methods.

**Weaknesses:**

* The experimental results listed in this paper are confusing and do not align well. The effectiveness is not very well proved.

* This paper proposes **TLSQ**, which is quite similar to TDQ in [1]. TDQ is applicable to diffusion models with both continuous time and arbitrary discrete time steps . The paper should clarify the number of time steps used in TLSQ and discuss the settings.

[1] Temporal dynamic quantization for diffusion models.

**Questions:**

* In Table 2, the FID score of W4A4 model (6.17) is much lower than FP model (11.28) and W8A8 model (11.38) is comparable to FP model. Is there any possible explanation for that? And in the paper "High-Resolution Image Synthesis with Latent Diffusion Models", FID score of conditional generation on ImageNet 256x256 is 3.60, which is much lower than 11.28, why is there a gap? In the Appendix.A Table.A, the unconditional image generation on LSUN dataset, the FID score of W4A4 model is much worse than the FP model, how to explain the gap in these two set of experiments?

* Table 3 shows the ablation study results. Does the **QALoRA** use LSQ algorithm to fine-tune the low rank adapter parameters and quantization step-size parameters?

* Are there any results on using LSQ method on quantized diffusion models on dataset other than Cifar10?

* In **Data-free fine-tuning for diffusion models** part, is $\mathbf{x}_t$ in Eq(7) sampled from Gaussion noise with an FP model?

* In **Variation of activation distribution across steps** part, it proposes to assign a separate step size parameter for activation quantization in each denoising time step, and the results shown in Table 2 are obtained from 20-step sampling. Is the total time steps fixed to 20 for the fine-tuning. Is the data-free fine-tuning in Eq(7) fixed for 20 steps?

* In Sec3.2 Eq(3), the quantization scheme has three parameters, $l, u, s$, are they all trainable? If so, is it optimized with LSQ [2] or LSQ+ [3] algorithm?


[1] High-Resolution Image Synthesis with Latent Diffusion Models.

[2] Learned step size quantization.

[3] LSQ+: Improving low-bit quantization through learnable offsets and better initialization.

---

> ### Author Response · Authors · 2023-11-17
> **Rebuttal by Authors**
>
> Thanks to the reviewer for the valuable comments.
>
> **Q1: The experimental results listed in this paper are confusing and do not align well. In Table 2, the FID score of W4A4 model is much lower than FP model. While in Table A, the FID score of W4A4 model is much worse than the FP model. How to explain the gap in these two sets of experiments?**
> There might be a misunderstanding of our results. To clarify, **while FID provides an informative metric, it might not holistically capture the improved image quality**. Therefore, referring to Table 2 in the paper, we thoroughly evaluate EfficientDM's performance by reporting the results of four standard metrics: IS, FID, sFID, and Precision. It's crucial to emphasize the **comprehensive integration of these metrics** when evaluating the model, rather than relying solely on any individual metric. Through a thorough evaluation with these metrics, FP models consistently outperform W4A4 models. Moreover, we have provided visualizations in Figures D-J in the supplementary material, convincingly showcasing the quality of images generated by EfficientDM.
>
> **Q2: Why is there a gap between the FID results reported in this paper and LDM [i]?**
> The settings are different. LDM paper uses 250 DDIM steps to obtain an FID of 3.60, while we use 20 DDIM steps to obtain an FID of 11.28.
>
> **Q3: The paper should clarify the number of time steps used in TLSQ and discuss the settings.**
> In practice, we set the number of time steps as 100 during fine-tuning and interpolate the learned temporal quantization scales when sampling with fewer time steps. We include the settings in Section 5.1 in the revised paper.
>
> **Q4: Does the QALoRA use LSQ algorithm to fine-tune the low rank adapter parameters and quantization step-size parameters?**
> Yes, the weight quantization step-size parameters are optimized during the fine-tuning process.
>
> **Q5: Additional experimental results of using LSQ [ii] method on quantized diffusion models.**
> As referred to Table A in the supplementary material, we conduct experiments using LSQ method on LSUN dataset. Despite consuming $15.8\times$ less time, our approach surpasses LSQ at W8A8 bit-width, demonstrating superior FID performance.  At W6A6 bit-width, our approach achieves an FID that is only 0.07 higher than LSQ, striking a better balance between performance and efficiency.
>
> **Q6: Is $\mathbf{x}_t$ in Eq. (7) sampled from Gaussian noise with an FP model?** Yes. Please refer to Q2 in the general response. The input data $\mathbf{x}_t$ is derived from denoising random Gaussian noise $\mathbf{x}_T \sim \mathcal{N}(0,1)$ with the FP model through $T-t$ iterations. We have revised the explanation of Eq. (7) in the revised paper.
>
> **Q7: Is the data-free fine-tuning in Eq. (7) fixed for 20 steps?**
> As referred to Q2, we set the denoising step for fine-tuning as 100 in practice and interpolate the learned scales for sampling with fewer steps.
>
> **Q8: In Eq. (3), the quantization scheme has three parameters, are they all trainable?**
> No. Only quantization scale $s$ is trainable and optimized with LSQ algorithm, while $l$ and $u$ are determined by the target bit-width. We have revised it in the revised paper.
>
> [i] Rombach, Robin, et al. ``High-resolution image synthesis with latent diffusion models." CVPR, 2022.
>
> [ii] Esser, Steven K., et al. ``Learned step size quantization." ICLR 2020.

---

> ### Comment · Reviewer_ec6g · 2023-11-22
>
> Thanks for the authors' response. It well addresses my concerns.

---

> > ### Author Response · Authors · 2023-11-23
> > **Appreciation for Your Valuable Feedback**
> >
> > Dear Reviewer ec6g,
> >
> > Thank you for your feedback. We truly appreciate your careful consideration of our responses.
> >
> > Best regards,
> >
> > Authors of #3007.

---

> > > ### Comment · Reviewer_ec6g · 2023-11-23
> > >
> > > I have another question about the **Variation of activation distribution across steps** part. Since the training is on 100-step DDIM,  if there are 100 the activation quantizer step-size parameters, how is the training performed? I think a lot of step-size parameters will have 0 gradients in one training iteration. Did you do something to alleviate it?

---

> ### Author Response · Authors · 2023-11-23
> **Response to Reviewer ec6g**
>
> Dear Reviewer ec6g,
>
> As referred to "Variation of activation distribution across steps" part of the paper, we allocate temporal step-size parameters for activations and optimize them individually for each step. Therefore, during each training iteration, which corresponds to a single denoising step, only the step-size parameters associated with this step are updated, while other step sizes have zero gradients because they are not involved in forward propagation. **After a complete denoising process spanning all denoising steps, all temporal step-size parameters undergo optimization.**
>
> Best regards,
>
> Authors of #3007.

---

> > ### Comment · Reviewer_ec6g · 2023-11-23
> >
> > If an optimizer like Adam is applied, the momentum statistics will be affected by the zero gradients. Didn't it influence the training procedure?

---

> ### Author Response · Authors · 2023-11-23
> **Response to Reviewer ec6g**
>
> Dear Reviewer ec6g,
>
> In our method, during each training iteration corresponding to a single denoising step, **only the step-size parameters associated with this denoising step are updated**. To achieve this, we store a copy of the Adam optimizer's statistics for all other step sizes in a separate memory space and then temporarily zero them out to ensure they remain unaffected during the update process. For the step size parameters associated with this step, their statistics will be loaded back into the optimizer.
>
> Best regards,
>
> Authors of #3007.

---

### Author Response · Authors · 2023-11-17
**General Response**

We thank all reviewers for their valuable feedback. Overall, our work has been well recognized as:

* "The idea is novel" (Reviewer 5Wot)
* "Achieving QAT-level performance with PTQ-level efficiency is significant and promising" (Reviewers 5Wot)
* "It is the first to achieve very good image generation performance with W4A4 diffusion models" (Reviewer ec6g)
* "It demonstrates promising and strong performance" (Reviewers BLfX and SYoS)

We have summarized and addressed the main concerns as follows:

**Q1: The differences between the proposed TLSQ and TDQ [i].**

As mentioned in the "Related Work" section and the "Variation of activation distribution across steps" part of the paper, TDQ incorporates additional MLP modules for each quantized layer to approximate quantization scales. This approach entails training these extra MLP modules, leading to a significant surge in training expenses. Conversely, TLSQ introduces a limited number of adjustable quantization scales per layer, making TLSQ more computationally efficient than TDQ. Empirically, TLSQ achieves a 20.9% faster training iteration speed compared to TDQ over DDIM model on CIFAR-10 dataset.

**Q2: How to get $\mathbf{x}_t$ in Eq. (7)?**

The input data $\mathbf{x}_t$ is derived from denoising random Gaussian noise $\mathbf{x}_T \sim \mathcal{N}(0,1)$ with the FP model through $T-t$ iterations. We have revised the explanation of Eq. (7) in the revised paper.

[i] So J, Lee J, Ahn D, et al. "Temporal Dynamic Quantization for Diffusion Models." NeurIPS, 2023.

---

### Meta-Review · Area_Chair_mHjD · 2023-12-11

**Metareview:**

This paper presents a quantization method for diffusion models, which pushes the quantization of diffusion models to an impressive level. Reviewer are convinced by its effectiveness. Please include the clarifications requested by reviewers in the final version of the paper.

**Justification For Why Not Higher Score:**

Some essential technical details are missing

**Justification For Why Not Lower Score:**

This is a good paper with solid contribution and impressive results.

---

### Decision · Program_Chairs · 2024-01-16

Accept (spotlight)